# Breast Cancer Care Pathways for Women with Preexisting Severe Mental Disorders: Evidence of Disparities in France?

**DOI:** 10.3390/jcm12020412

**Published:** 2023-01-04

**Authors:** Anna-Veera Seppänen, Fabien Daniel, Sophie Houzard, Christine Le Bihan, Magali Coldefy, Coralie Gandré

**Affiliations:** 1Institut de Recherche et Documentation en Economie de la Santé (IRDES), 75019 Paris, France; 2Institut National du Cancer (French National Cancer Institute—INCa), 92513 Boulogne-Billancourt, France; 3AP-HP, Hôpital Universitaire Robert Debré, 75019 Paris, France

**Keywords:** breast cancer, mental disorders, care pathways, healthcare disparities, health services research, administrative claims, healthcare, France

## Abstract

The excess cancer mortality in persons with severe mental illness (SMI) has been well documented, and research suggests that it may be influenced by care-related factors. Our objective was to assess breast cancer care pathways in women with SMI in France, using an exhaustive population-based data-linkage study with a matched case-control design. The cases were 1346 women with incident breast cancer in 2013/2014 and preexisting SMI who were matched with three controls without SMI presenting similar demographics, initial breast cancer type, and year of incidence. We compared cancer care pathways and their quality for cases and controls, using a consensual set of indicators covering diagnosis, treatment, follow-up, and mortality (until 2017). After adjusting for covariates, cases had lower odds to undergo the main diagnostic tests, lumpectomy, adjuvant chemotherapy, and radiotherapy, as well as hormone therapy, but higher odds for mastectomy. Suboptimal quality in cancer pathways was observed for both groups, but to a higher extent for cases, especially for not receiving timely care after diagnosis and post-treatment follow-up. Breast cancer mortality, considering competing risks of deaths, was significantly elevated in women with SMI. These findings highlight disparities in cancer care pathways for individuals with SMI, as well as specific aspects of the care continuum which could benefit from targeted actions to reach equity of outcomes.

## 1. Introduction

Severe mental illnesses (SMIs) include a range of chronic and disabling conditions, such as bipolar and psychotic disorders, which often manifest in recurring episodes that cause severe impairment, notably by limiting functional capacities and social skills [1]. The excess mortality of people living with SMI has been known for decades [2,3,4], but it is persisting over time, or even worsening, as they have not experienced the gradual increase in life expectancy seen in the general population [5,6,7]. This health inequality is thought to result from a combination of factors, including shared risk factors for mental and somatic disorders, drug-related iatrogenesis, differences in pain perception linked to SMI and their treatment, and economic difficulties and living conditions (such as social isolation) that are unfavorable to health and likely to limit individual healthcare-seeking, as well as factors linked to the health system. These factors can be both organizational (such as compartmentalization of general and specialized care) and behavioral (such as a tendency among some health professionals to attribute somatic complaints of individuals living with an SMI to their mental disorder, which is known as ‘diagnostic overshadowing’) [8,9].

Similar to the rest of the population, the primary causes of death in persons with SMI include cardiovascular disorders and cancer, for which an excess mortality ratio of more than two is consistently found in the literature for the SMI group [10,11,12,13]. Engagement in risky health behaviors is often accused of increasing the odds of developing and dying of cancer in this population, but the scientific literature provides conflicting results on this matter. Some publications report higher overall cancer odds [14,15,16,17], while others have found a lower or similar incidence in persons with SMI compared to the general population [18,19]. Caution is needed when interpreting these findings, as there are age-related differences and variations with the type of cancer considered [17,19,20], but they suggest that factors beyond risky health behaviors, intervening after the onset of the disease, such as at the time of access to the health system or of care delivery, may play a significant role in the excess mortality by cancer demonstrated in the SMI population.

Previous work focusing on specific phases of cancer care in this vulnerable group has consistently highlighted less access to cancer screening [21,22] and lower care intensity than for the general population [23,24,25,26,27]. Many of the factors that contribute to the excess mortality in persons with SMI are therefore likely to stem from institutional and policy-level issues in care pathways, making this excess mortality a question of human rights and equity. However, despite the relevance of documenting cancer care pathways in individuals with SMI, research on this topic remains scarce. To the best of our knowledge, only a few studies have aimed at providing a global overview of cancer care pathways for people with SMI. A national study carried out in Japan, focusing on gastrointestinal cancer, found that patients with both cancer and schizophrenia were more likely to have advanced cancer upon admission than those without SMI, were less likely to receive surgical or endoscopic treatment after adjusting for cancer stage, and had higher in-hospital mortality within 30 days [28]. Another study, also conducted in Japan, but on breast cancer, found that patients with schizophrenia were less likely to receive chemotherapy or the recommended treatment compared to cancer patients without schizophrenia [29], while a study carried out at a large scale in Finland for the same type of cancer found a lower access to radiotherapy specifically for women with SMI [30]. In Canada, individuals with SMI and colorectal cancer were less likely to receive potentially curative surgical resection, as well as adjuvant radiation or chemotherapy [31]. Research carried out in one Australian state for different cancer sites showed similar results, with a higher proportion of cancer with metastases at diagnosis, a reduced likelihood of surgery and radiotherapy, and fewer chemotherapy sessions overall among the population with SMI [11]. Despite a relative consistency in these first findings, there is still a dearth of research resorting to linked data between community and hospital care to comprehensively document cancer care pathways for people with SMI in different national settings, using large sample sizes, consensual indicators of care quality within the context of clinical guidelines, and a control group, as recently advocated [31,32,33,34].

In this context, our objectives were to assess cancer care pathways from diagnosis to death for individuals with a preexisting SMI compared to individuals without, at a national scale in France, using breast cancer—one of the most prevalent cancers—as an illustrative example.

## 2. Materials and Methods

### 2.1. Conceptual Framework

Based on the scientific literature, healthcare disparities are defined in our research as differences in access to and quality of healthcare which are not due to clinical appropriateness and patient needs [35]. Furthermore, our theoretical framework conceptualizes that if differences in cancer care pathways are found for individuals with SMI—in relation to guidelines and after taking into account socioeconomic vulnerabilities and individual clinical needs (for example linked to the type of cancer at presentation or medical comorbidities)—they represent care disparities specific to people with SMI. While some previous research considers that adjusting for other vulnerabilities may lead to underestimating effects [31], our theoretical framework provides conservative estimates that enable us to isolate the associations between preexisting SMI and differences in cancer care pathways and can help develop more tailored interventions. This choice is in line with the most recent 10-year French National Cancer Control Strategy, which highlights the need to provide evidence-based findings on health inequalities in cancer care by transitioning from a general to a targeted approach focusing on the specificities of some vulnerable population groups [36].

### 2.2. Setting

In France, the significant excess mortality of people living with a mental disorder has only recently been documented exhaustively, thanks to new data-linkage opportunities, confirming the role of cancer as the primary cause of death in the French SMI population [10]. Our focus on breast cancer is driven by its epidemiological weight in France: for women, it is the most common form of cancer (33% of all cancer cases in 2018) and the most common cause of death from cancer (causing 12,100 deaths in 2018) [37].

The French health system is characterized by limited collaboration between hospital, ambulatory, and social care providers, as well as between somatic and psychiatric care, which can result in fragmented care and breaks in care continuity [38]. However, positive features of this system include low out-of-pocket costs [39] resulting from a universal statutory health insurance (SHI) system and a high complementary health insurance coverage. Persons with the lowest incomes can receive state-subsidized complementary health insurance (*Couverture maladie universelle complémentaire*, CMU-C) or, when not eligible, allowances to help purchase a complementary health insurance (*Aide à l’acquisition d’une complémentaire santé*, ACS)—two schemes that were combined under one single scheme, the *Complémentaire santé solidaire* (C2S) in 2019 [40]. Persons with certain chronic conditions, including SMI and cancer, can be covered by long-term illness (LTI) schemes, limiting co-payments for healthcare.

A mammogram is offered free of charge every two years to all women aged between 50 and 74 years, as part of the national breast cancer screening program, or whenever prescribed by a physician. Any further diagnostic exams are reimbursed by the SHI, and in the case of cancer, the primary care provider refers the patient to a multidisciplinary team specialized in cancer [41]. Cancer care is provided in both (profit or non-profit) private and (general or teaching) public hospitals, as well as in highly specialized non-profit cancer centers (*Centres de lutte contre le cancer*, CLCC), of which there is one in each region. Hospitals need to perform a minimum amount of cancer interventions per year (30 surgeries in the case of breast cancer) in order to be authorized to provide cancer surgery, chemotherapy, and radiotherapy [42]. Radiotherapy is mainly provided in private clinics, whereas approximately half of chemotherapy is provided in public hospitals. Although relatively few in number, the CLCC performed 14% of all chemotherapy and 21% of all radiotherapy sessions for all cancer sites in 2019 [43].

### 2.3. Study Design and Data Source

This research relies on a population-based matched case-control study, using linked data from the national health claims database (*Système national des données de santé,* SNDS), covering almost 100% of the French population. The SNDS contains all healthcare acts reimbursed by the SHI, including drug prescriptions, outpatient medical procedures, and visits, as well as stays in private and public hospitals. It also provides data on time of death and its causes (for individuals covered by the SHI general scheme—*régime général*—which includes around 90% of the French population) [44] and certain sociodemographic indicators of patients, such as age, sex, residential zip code, and enrollment to a state-subsidized complementary health insurance scheme (CMU-C or ACS) [45]. All of these data are linked with a unique pseudonymized patient identifier [46].

As the SNDS does not contain clinical examination or test results, patients with chronic conditions, covered by the SHI general scheme, are identified via a mapping tool (*Cartographie médicalisée*). This tool combines information over the past five years on causes of hospitalizations, conditions leading to entry in the LTI scheme, drug prescriptions, and medical procedures in order to identify individuals suffering from specific chronic conditions, including SMI and cancer [47].

According to French law, ethical approval and informed consent from patients is not required for this study; the research team has access to fully pseudonymized SNDS data under strict data-protection protocols [48].

### 2.4. Study Population

Our initial study population comprised female adult patients with an incident treated breast cancer in either 2013 or 2014 (International classification of diseases, tenth revision, ICD-10 codes C50, D05, and D486) included in the SNDS mapping tool. Incidence was defined by an absence of hospitalizations or inclusion in the LTI scheme for breast cancer in the previous five years and an absence of specific drug treatment in the previous two years. We excluded women treated for non-mammary cancers, either concomitant or in the previous year, to avoid heterogeneity in care pathways due to treatment of other cancers with different care recommendations. The incidence years were chosen to enable sufficient follow-up for mortality data, as causes of deaths are only available in the SNDS database until 2017 so far. Cancer incidence was considered over two years to obtain a sufficient sample size of patients with both SMI and breast cancer. The initial cancer type (in particular, ductal carcinoma in situ, invasive cancer with or without lymph node involvement, or metastasis) was determined according to the ICD-10 causes of hospitalization one year before and six months after the first treatment for breast cancer. To consider breast cancer care pathways with consistent guidelines, we then specifically focused our main analysis on women with non-metastatic invasive (NMI) breast cancer.

Patients with SMI were women with a preexisting SMI one year prior to breast cancer incidence consideration to improve the likelihood that the mental illness was not related to a new cancer diagnosis. In addition, we only considered the most severe mental disorders according to the literature [34,49,50] and those that are less likely to be triggered by the diagnosis of a severe somatic condition (compared to, for instance, depressive episodes). We therefore considered psychotic disorders (ICD-10 codes F20-F29), as well as mania and bipolar affective disorders (ICD-10 codes F30 and F31). Patients with SMI the year prior to breast cancer diagnosis were identified in the SNDS mapping tool based on a hospitalization for one of these disorders in the last two years, a hospitalization for one of these disorders in the last five years if it was also combined with a recurrent psychotic drug treatment, or inclusion in the LTI scheme for one of these disorders [51]. Individuals not identified with a SMI in the year prior to breast cancer diagnosis but identified with such a disorder in 2013 or 2014, and with a recurrent psychotic drug treatment, were classified in the SMI group, considering the cyclic nature of mental disorders.

### 2.5. Matching between Cases and Controls

A case-control study was carried out to identify the specific effect of SMI on cancer care pathways. Included cases were all women with a preexisting SMI and incident breast cancer (see definition above), restrained to cases of NMI breast cancer, for which specific consensual relevant indicators of the quality of care pathways are available (see Section 2.6). To study comparable groups in terms of demographics and cancer type, we used an exact matching method with replacement to select three controls (without preexisting SMI) per case and to obtain a balanced number of matched individuals. The following matching criteria were used: age (categorized in five-year groups), local county (*département*) of residence, calendar year of first treatment for breast cancer (2013 or 2014), and initial type of breast cancer (NMI breast cancer with lymph node involvement or NMI breast cancer without lymph node involvement). Women who developed an SMI during the year of cancer incidence were excluded from the pool of potential controls.

We discarded cases for whom three exact controls could not be found. They represented a minority of all cases, and we described their main characteristics in comparison to the matched cases.

### 2.6. Indicators of Cancer Care Pathways and Mortality

To characterize NMI breast cancer care, we identified which of the main diagnosis exams (mammogram, biopsy, echography, and magnetic resonance imaging (MRI)) and treatments (neoadjuvant chemotherapy, lumpectomy, mastectomy, hormone therapy, adjuvant chemotherapy—including targeted therapy—and radiotherapy) our study population had received. We also described their combination, which was characterized in line with previous research on breast cancer care in the SMI population [30].

To compare the quality of their care pathways, we used a set of consensual and complementary indicators for adult women with incident-treated NMI breast cancer. They were developed by the French national cancer institute (*Institut national du cancer*, INCa), in partnership with the French national health authority (*Haute autorité de santé*, HAS), based on a review of the literature of good clinical practice and care pathway recommendations, and by consulting experts, scientific societies and cancer survivors. The indicators cover diagnosis, treatment, and follow-up of breast cancer. For each indicator, a target threshold (representing the ideal share of women for which the indicator should be met) and an alert threshold (representing the share of women under or above which there should be concerns regarding the quality of the care provided) have been previously defined based on expert consensus [52,53]. The indicators were first programmed on the French cancer cohort by INCa before being adapted and applied to our study population.

Finally, we assessed breast-cancer-specific mortality, i.e., deaths with an ICD-10 code relating to breast cancer (codes C50, D05, or D486) recorded as the underlying cause of death. Mortality follow-up was only conducted for individuals with linked death-certificate data. This follow-up started at the date of first treatment for breast cancer and ended at the earliest date among date of death, date of last known care consumption in the SHI general scheme (for individuals who left this scheme on the study period), or 31 December 2017 (latest availability of data for causes of death at the time of the study).

### 2.7. Adjustment for Covariates

In addition to whether women had a preexisting SMI, we adjusted for other variables that may influence cancer care pathways, including socioeconomic status, overall health state, and characteristics of the hospital where care was received, which may influence treatment choice and mortality [54,55,56,57].

Socioeconomic variables included an indicator at the individual level relating to inclusion in the CMU-C or ACS schemes helping to cover healthcare costs for individuals with the lowest incomes (see Section 2.2) and the quintile of a community-level deprivation index calculated at the patient’s residential zip code [58]. This index, named FDep, was specifically developed for the French context and considers the median household income, the percentage of high school graduates in the population aged 15 years and older, the percentage of blue-collar workers in the active population, and the unemployment rate. Quintiles of this index, computed using the French general population as a reference, range from least deprived (Q1) to most deprived (Q5).

Overall health state was considered by calculating a synthetic comorbidity index predicting mortality, specifically developed for the SNDS data: the mortality-related morbidity index (MRMI) [59]. This index was adapted to our study population by not including SMI or cancer among comorbidities (modified MRMI), as they were already considered in our analysis. While the use of this index is optimal within a population that has not been selected on a given condition, it still demonstrates a higher performance than other indexes for individuals with specific disorders [60].

Finally, to account for the characteristics of the hospital where care was received, we used a categorical variable indicating the type of hospital where each woman received her first breast cancer treatment (either a public general hospital, a public teaching hospital, a CLCC, a non-specialized private non-profit hospital, or a non-specialized private-for-profit hospital).

### 2.8. Analysis

We first described and compared the characteristics of the entire study population with breast cancer for women with and without preexisting SMI in terms of main demographic, socioeconomic, and clinical characteristics. We also compared these characteristics for matched and unmatched cases. Chi-square tests were used for categorical variables, and Wilcoxon tests were used for continuous ones.

We then described the general cancer care received by matched cases and controls with NMI breast cancer, calculating crude rates for each type of cancer care received and each quality indicator, and compared them by using univariate conditional regressions. To adjust for covariates, we carried out multivariable logistic regression models with a binomial response distribution, a log link function, and a repeated statement for matched cases and controls for binary indicators of care received or quality. For count indicators, which did not present over-dispersion, we conducted Poisson regressions, which also included a repeated statement for matched cases and controls.

To examine mortality outcomes, we carried out a competing risk analysis, using a subdistribution hazard model, considering death from breast cancer as the primary event and deaths from other causes as the competing risk, while adjusting for covariates. Survival/follow-up time was defined as the interval between the date of first contact with the health system for breast cancer and the date of breast cancer death or censoring. Comparison of cumulative incidence curves for cases and controls was based on Gray’s test [61]. The proportional hazards assumption was evaluated by checking for non-significant relationships between residuals and time.

For each type of care and quality indicator and for the competing risk analysis, covariates (see Section 2.7) were included in the multivariable model when they were significant in the univariate analysis at the 0.10 statistical significance level, after testing for correlation among these variables and checking that the final model chosen allowed minimization of the quasi-likelihood information criterion (QICu) or Bayesian information criterion (BIC). For indicators relating to diagnostic tests, the type of hospital where the first breast cancer treatment was received was not controlled for. In multivariable analyses, the magnitude of associations was measured by adjusted odds ratios (AORs) or hazard ratios (AHRs) and their 95% confidence intervals (95% CI), with a two-sided *p*-value of 0.05 or less denoting statistical significance.

Three sensitivity analyses were carried out for the competing risk analysis of death: (1) including deaths from any cancer site as the primary event (to account for potential deaths by metastases from the initial breast cancer); (2) including deaths with breast cancer recorded as an associated cause of death among primary events (to account for potential misclassification of the underlying cause in the death certificate and for complex deaths triggered by several causes—including breast cancer); and (3) adding indicators of the quality of breast cancer pathways that varied significantly between matched cases and controls as covariates to assess whether it impacted potential differences in breast cancer mortality across groups.

Analyses were mostly performed with the SAS EG software version 7.15 HF8 (SAS Institute Inc., SAS Campus Drive, Cary, NC, USA), but the competing risk analyses were carried out with the R software (version 4.1.2), using the R Studio interface (version 2022.02.2 + 485.pro2) and the package cmprsk (version 2.2-7).

## 3. Results

### 3.1. General Characteristics of the Study Population

Overall, 97,760 women met our inclusion criteria (see flowchart, Appendix A). A total of 48,992 had an incident (treated) breast cancer in 2013, and 48,768 in 2014. Among them, 1581 (2%) had a preexisting SMI (37.5% were diagnosed with a bipolar disorder, 56.1% with a psychotic disorder, and 6.4% with both types of disorders). They were characterized on average by a higher frequency of inclusion in schemes helping to cover healthcare costs for the most deprived individuals (CMU-C and ACS) and residency in more deprived areas compared to the women without SMI. Women with SMI were also statistically significantly less likely to have ductal carcinoma in situ and more likely to have metastatic breast cancer at presentation, even if the differences with women without SMI were rather minor, while their comorbidity index was higher on average (Table 1).

Three controls were found for 1346 (97%) of women with SMI and NMI breast cancer. A total of 263 (19.5%) had non-metastatic invasive breast cancer with lymph node involvement, and 1083 (80.5%) had non-metastatic invasive breast cancer without lymph node involvement. The characteristics of matched and unmatched cases are presented in the Appendix A. Matched cases were globally similar to the overall cohort of women with preexisting SMI and incident breast cancer. Unmatched cases were younger, more socioeconomically deprived, and with a higher proportion of non-metastatic invasive breast cancer with lymph node involvement than matched cases (Appendix A).

### 3.2. Global Cancer Care Received According to SMI Status

After adjustment, women with SMI were significantly less likely to undergo all main diagnostic tests (mammogram, breast biopsy, echography, and MRI) than women without SMI. Overall, they received a lower number of diagnostic tests and were less likely to receive the most recommended combination of tests (mammogram and breast biopsy) (AOR = 0.54, 95% CI: 0.46–0.64) (Table 2). Cases were also more likely to undergo mastectomy (AOR =1.38, 95% CI: 1.20–1.58) and significantly less likely to undergo lumpectomy (AOR = 0.82, 95% CI: 0.70–0.95) and to receive hormone therapy (AOR = 0.86, 95% CI: 0.75–0.99), adjuvant chemotherapy (AOR = 0.80, 95% CI: 0.70–0.91) and radiotherapy (AOR = 0.87, 95% CI: 0.75–0.98) (Table 2). Cases were more likely to have received operative treatment only (AOR = 1.47, 95% CI: 1.13–1.91) and less likely to have received a combination of operative and radiotherapy and chemotherapy (or hormone therapy) treatment (AOR = 0.84, 95% CI: 0.74–0.96).

### 3.3. Indicators of the Quality of Cancer Care Pathways According to SMI Status

Regarding the indicators of breast cancer quality of care, the target threshold was not met for any of the indicators for women with SMI. Indicators for women with SMI were also not meeting the alert threshold in most cases, except for the proportion of women undergoing radiotherapy after breast-conserving surgery (Indicator 6) (92% vs. an alert threshold of <90%) and for the proportion of women not treated with neoadjuvant chemotherapy undergoing breast reintervention (Indicator 9) (17% vs. an alert threshold of >20%) (Table 3).

However, non-compliance with target and alert thresholds was also observed for controls, although to a lesser extent for all indicators. The differences between cases and controls were the most marked for process indicators of diagnosis leading to treatment, for which women with SMI were statistically less likely to receive care in accordance with guidelines (less likely to undergo their first treatment within six weeks post-mammogram (Indicator 1), to undergo their first treatment within four weeks post-biopsy (Indicator 3), and to undergo biopsy prior to first treatment (Indicator 4)), even after adjustment for covariates (AOR between 0.65 and 0.78) (Table 3). Cases without lymph node involvement were also statistically less likely to undergo sentinel lymph node excision without axillary dissection (Indicator 5) (AOR = 0.82, 95% CI: 0.67–0.99). Besides these indicators linked to diagnosis, cases were statistically less likely to have had a follow-up mammogram post-treatment (Indicator 10) (AOR = 0.84, 95% CI: 0.73–0.97) (Table 3).

### 3.4. Mortality Outcomes

A total of 424 deaths occurred during the follow-up period: 180 in women with SMI (13% of cases: death for 13% of those with NMI breast cancer without lymph node involvement and 17% of those with NMI breast cancer with lymph node involvement) and 244 in women without SMI (6% of controls: death for 6% of those with NMI breast cancer without lymph node involvement and 7% of those with NMI breast cancer with lymph node involvement). Data linkage with causes of death was possible for 90% of these deaths (n = 380). The median follow-up time reached four years among survivors or women with censored data. Median survival time reached two years among both women who died from breast cancer and those who died from other causes, with little differences between cases and controls. Of those who died, 45% of women with SMI died from breast cancer as the underlying cause compared to 58% of controls. The two other most frequent underlying causes of deaths were external causes and cardiovascular disorders for cases and other cancers and cardiovascular disorders for controls. In the competing risk analysis, the incidence of breast-cancer-specific mortality was greater in cases compared to controls at all time points, as was the incidence of mortality by other causes (Figure 1).

After adjusting for covariates, the increased breast-cancer-specific mortality risk in women with SMI compared to their control remained (aHR = 1.39, 95% CI: 1.03–1.87) (Table 4). This finding was consistent when considering deaths from any cancer site as the primary event and deaths with breast cancer recorded as an associated cause of death among primary events, but in these two cases, the mortality risk increased for women with SMI (Table 4). On the contrary, when adding separately each indicator of the quality of breast cancer pathways that varied significantly between the population with or without SMI (Indicators 1, 3, and 4 relating to diagnosis process; Indicator 5 relating to process between diagnosis and treatment; and Indicator 10 relating to follow-up—see Table 3) as covariates, the increased breast cancer specific mortality risk for women with SMI compared to their control was no longer significant (except for Indicator 10).

## 4. Discussion

Our research demonstrates that women with a preexisting SMI who were treated for NMI breast cancer in France received a lower number of diagnostic tests overall, being in particular less likely to receive the most recommended combination of tests than matched controls without a preexisting SMI, presenting similar demographic, socioeconomic, and clinical characteristics. Even when they did receive recommended tests, they were less likely to undergo their first treatment within six weeks post-mammogram, to undergo their first treatment within four weeks post-biopsy, and to undergo biopsy prior to first treatment in line with guidelines, underscoring issues in timeliness between diagnosis and treatment. In addition, they received cancer treatment of lower intensity, as they were less likely to receive adjuvant chemotherapy and radiotherapy, as well as hormone therapy, and more likely to receive operative care only in comparison to controls presenting the same type of breast cancer and even after adjustment on their overall health. Similarly, women with SMI were significantly less likely to receive adequate post-treatment follow-up. Overall, none of the consensual target thresholds were met for the indicators of the quality of breast cancer care pathways in the SMI population. However, this was also the case for women without a preexisting SMI, which corroborates results from studies using recent data on the overall population treated for breast cancer in France [53]. The odds for meeting care-quality targets were lower for women with SMI for all indicators, but statistically significant differences between cases and controls were only found for indicators of diagnosis process, process between diagnosis and treatment, and follow-up post-treatment. Finally, women with SMI presented a higher breast cancer-specific mortality risk than the matched controls, after taking into account competing risks of deaths and comorbidities. This increased risk was no longer significant in an exploratory sensitivity analysis when adjusted for indicators of the quality of breast cancer pathways, relating to diagnosis process and process between diagnosis and treatment, that varied significantly between the population with or without SMI; this should, however, be complemented by further research to conclude on causality.

Our findings are largely consistent with the scarce existing works in the literature. In Finland, women with SMI were previously found to be more often diagnosed at the metastasized stage [30], similarly to what was observed in our study population. Research carried out in other national settings also demonstrated higher rates of mastectomy and less intensive care, with, in particular, lower access to radiotherapy, for women with breast cancer and SMI [30,33]. Regarding mortality outcomes, previous studies have systematically found an increased risk of breast cancer mortality in individuals with a preexisting SMI compared to those without and consistently found findings in the same order of magnitude as ours when considering competing risks of deaths [30,32]. Our results are also in line with research on other cancer sites that has also found a lower care intensity and likelihood of receiving guideline-recommended treatment for individuals with SMI, as well as finding a higher risk of mortality [28,31].

Our findings underscore that breast cancer care pathways are not optimal for women with SMI in France, and less so than for comparable women without SMI on several aspects of the care continuum. Diagnostic processes differ between the two groups with more frequent metastasized cancer at presentation, less receipt of the main diagnostic tests, and less timely treatment in the SMI population, with a potential impact on their chances of survival. Exploratory discussions with clinicians allow us to draw several hypotheses to explain these results. They could reflect more frequent receipt of non-standard diagnostic testing (such as computed tomography scans), which may be linked to a higher rate of fortuitous cancer discovery (and a lower participation in the national cancer screening program), in women with SMI. Similarly, they may also be having more surgeries initially thought to be for non-malignant health issues, inadequately carried out without biopsies, and finally leading to a cancer diagnosis. Our results could also partly reflect an adaptation to the complex needs of some patients with SMI for whom carrying out diagnostic exams under general anesthesia may be preferable, which could lead to an immediate surgical procedure after diagnosis, without biopsy, to limit the number of anesthetic procedures. Finally, they could reflect inappropriate medical checkups (for instance MRI or breast palpations only) leading to cancer treatment for the SMI population, which can be problematic in a context where overdiagnosis of tumors not needing treatment is increasingly being discussed [62]. In our research, women with SMI were also less likely to receive intensive cancer treatment (including adjuvant chemotherapy, radiotherapy, and hormone therapy) than similar controls (notably presenting the same type of breast cancer, living in the same local area, and after adjustment for the type of hospital providing the first breast cancer treatment). Several hypotheses can be drawn up to explain these findings. While radiotherapy is a mandatory component of treatment when the surgery is breast-conserving, it is not systematically indicated after mastectomy, an operation which is more frequent among women with SMI. This questions equity of care due to the more invasive nature of this procedure and the fact that it may be used to avoid dealing with radiotherapy appointments for women with SMI. Previous research has highlighted that concerns regarding compliance with treatment in specific patient groups may lead some healthcare professionals to avoid offering multiple sessions of chemotherapy and radiotherapy, or even hormone therapy, as a treatment option [33,55], which can constitute a potential prejudice towards these patients. It could also be linked to fears of neuropsychiatric symptoms secondary to the use of chemotherapy [63] or of interactions between anticancer drugs and psychotropic agents [64]. However, previous research suggests that not resorting to certain types of cancer treatment can sometimes result from patients’ choice. Instances where women with breast cancer and SMI have refused chemotherapy can be found in the literature, but this was also the case for some women without SMI and was explained by individual perceptions of risk and benefits [65]. While we do not have access to information on medical and personal decisions regarding treatment options, our results suggest that the French health system fails to respond to the complex needs of women with breast cancer and SMI. Further interviews with patients, their relatives, and healthcare professionals conducted in a qualitative component of the same research project may shed additional light on this issue. Finally, the lower access of women with SMI to the most intensive combination of cancer treatment could be explained by clinical factors which were not measurable in our research, such as grading, hormonal receptor status (but we matched cases and controls on age that strongly drives this status), biomarkers, or histological subtypes which are important for some treatment choices [66]. It should also be noted that new research findings are constantly emerging on the right intensity of treatment for each type of breast cancer [67,68]. In addition, the most significant differences in indicators of the quality of cancer care pathways between women with and without SMI were found for indicators linked to diagnosis and follow-up rather than treatment itself.

Our findings should be interpreted while considering some limitations. Previous research has notably found that the excess mortality from breast cancer in women with SMI can be partly explained by the cancer stage at diagnosis [30], as it is consistently found to be more advanced in people with SMI [69]. In our research, we were not able to assess cancer stage beyond identifying the initial cancer type (such as metastatic, ductal carcinoma in situ, or NMI breast cancer). Furthermore, while we were able to adjust our analysis on several socioeconomic characteristics, their availability remains limited in the SNDS data. For instance, the data do not include any indicator of social support or marital status, which are likely to play a role in cancer care pathways. The presence of caregivers or strong family support can indeed be expected to facilitate the timely care of patients in comparison to those who are secluded. Their role has been little addressed in the literature on SMI and cancer so far [70], and the qualitative component of our research project, which includes interviews with relatives and caregivers of patients with SMI, may shed additional light on this issue. Similarly, the data do not provide any information on SMI symptom levels and functioning, nor on adherence to pharmacological medications—that can be particularly complicated for psychotropic drugs [71], while there may be specific challenges associated with the care of patients with unstable or untreated psychiatric disorders compared to patients with stabilized SMI [33]. In addition, while we tried to provide a comprehensive picture of cancer care pathways, our research lacks some possible elements of these trajectories including access to clinical trials [72], breaks/disruptions in planned cancer treatment [33], pain management during cancer care [73], and the occurrence of multidisciplinary team meetings, which are not currently available at the national scale. Regarding our statistical analysis, despite a small number of cases for whom three controls could not be found, unmatched cases were more socioeconomically deprived than matched cases, and our findings may therefore not be fully generalizable to the most vulnerable women with SMI. While previous research has underscored that social inequities in cancer treatment are less significant for females than for males, it has also highlighted that they were stronger for the most treatable cancers, such as breast cancer [55]. Moreover, our findings on the lack of significant differences between cases and controls regarding certain indicators (notably quality of cancer care pathways in terms of treatment process and outcomes) should be interpreted with caution. While we estimate that our sample size was adequate to detect relative differences of at least 15% in indicators of treatment and mortality between women with and without SMI with an alpha risk of 0.05 and a power of 80% (two-sided test), our study was underpowered to detect smaller differences in quality indicators. Our results therefore only provide a conservative estimate. Additionally, despite matching patients on their local county of residence and adjusting on the type of hospital where the first breast cancer treatment was received, we were not able to fully account for potential geographical disparities in access to cancer care, nor were we able to account for potential correlations in the data of patients treated by the same care provider in our models, as some providers only cared for a limited number of patients from our study population, and some patients had several providers. Finally, in order to have a sufficient follow-up period for mortality outcomes, we assessed care pathways for breast cancers that were incident in 2013 and 2014, and such pathways may have evolved in recent years.

Our research presents several strengths addressing lacks in the previous literature. First, we provided an exhaustive and representative picture of breast cancer care pathways of women with SMI at a large scale, in comparison to women without such disorders, using comprehensive population-based linked administrative data (including healthcare utilization and time and causes of deaths) in a universal healthcare setting, which avoids selection, information, and recall bias. Second, we resorted to a closely comparable control group, in terms of demographic, socioeconomic, and clinical characteristics, while previous research often lacks a reference group or is limited to data collected in the general population. Our method further enabled the isolation of the specific association between SMI and breast cancer care pathways, considering the multiple vulnerabilities faced by individuals with SMI. Finally, we relied on a complementary set of outcome variables covering diagnosis, treatment, follow-up, and mortality. They included highly consensual indicators of the quality of cancer care pathways, taking into account the particularities of a specific cancer site and specifically developed for the SNDS data [53]. We used information on causes of deaths, for which data linkage was much higher than in previous research [32], and that was analyzed by considering competing causes of deaths. To the best of our knowledge, there is no equivalent in the current literature exploring the link between SMI and cancer care.

Our results highlight opportunities to intervene throughout the continuum of cancer care to promote quality of breast cancer care pathways and equity of outcomes for women with SMI. First, the higher share of metastasized cancer at presentation, the lower occurrence of recommended diagnostic tests for these women, and the inadequate timeliness between diagnosis and treatment call for primary care physicians to ensure that cancer screening is adequately prescribed and promoted in the SMI population, that tests received are in line with guidelines, and that they are quickly followed by treatment. This could be encouraged by increasing awareness of primary care providers through targeted initial or continuous training. Our findings on less intensive treatment and overall lower odds for meeting care quality targets for women with SMI call for developing more collaboration between primary care professionals, oncologists, and mental health providers, such as public ambulatory care centers, which follow most of the SMI population in France, throughout the cancer care continuum. This kind of collaboration could be materialized through regular multidisciplinary meetings, for instance organized by hospital psycho-oncological support teams. The mental healthcare sector could also play a proactive role in accompanying its patients towards cancer screening, in informing somatic healthcare professionals about the specificities of persons with SMI, and in facilitating compliance with cancer treatment. While such initiatives are emerging locally in France, they could be made nationally available, supported by recent reforms, such as a new policy that has recently created local territorial networks of diverse professionals (*projets territoriaux de santé mentale*) to ensure meeting the mental healthcare needs of the population as a whole but also to provide adequate somatic care and social integration for people with SMI [74]. The concurrent emergence of advanced nurse practitioners, with specializations including oncology and mental health [75,76], could also offer some opportunities for further care management of comorbid SMI and cancer. Combined and multifaceted interventions indeed remain necessary to reach equity of outcomes, in particular in terms of cancer specific mortality.

Our work highlights several avenues for future research. This study is part of a larger research project relying on mixed methods to highlight potential issues in cancer care pathways for individuals with SMI. If measuring the extent of disparities in comparison to the general population is essential, it is no less important to characterize their causal mechanisms and understand whether they result from individual, organizational or structural factors. This will be facilitated by qualitative semi-structured interviews carried out with patients, their relatives, and health professionals. They will also help identifying potential contrasted realities in cancer care trajectories, which could be concurrently explored quantitatively, using, for instance, clustering methods. As we primarily aimed at describing cancer care pathways for people with SMI at a population level, rather than at explaining between-group differences, our first findings indeed provide informative but average conclusions that may hide variations (for example according to the severity of the SMI symptomatology). Our research will also be replicated on other cancer sites for which consensual indicators of the quality of care pathways are available at the national scale to see whether consistent conclusions can be drawn. Finally, while our findings suggest that less optimal cancer care pathways, notably in terms of diagnosis processes, could be related to worse mortality outcomes for people with SMI, additional research is needed to determine their formal links and potential mediating or confounding factors.

## 5. Conclusions

Our research has highlighted disparities in the breast cancer care pathways of women with SMI, in comparison to controls without SMI, at a national scale in France. Providing data on care disparities experienced by this vulnerable population, which has been neglected in health-services research focusing on care inequities, is a critical first step towards action. Additional research on causal mechanisms will help inform the development of system-level multifaceted interventions, with an understanding that the complexity related to SMI requires special consideration and that providing increased quality of care for this population group has the potential to make up for some of the structural health inequities they face throughout their life.

## Figures and Tables

**Figure 1 jcm-12-00412-f001:**
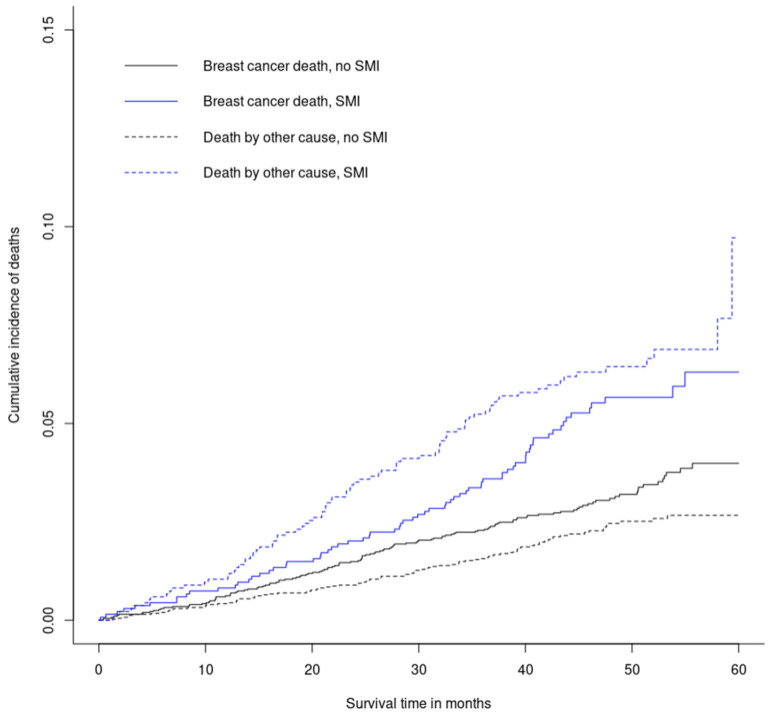
Cumulative incidence of breast cancer deaths (solid lines) and deaths by other causes (dashed lines), considered as competing risks, for women with and without SMI *. * Cumulative incidence curves for breast cancer specific mortality and mortality by other causes were highly statistically different (*p* < 0.001) between women with and without SMI.

**Table 1 jcm-12-00412-t001:** Main characteristics of women with incident breast cancer (all types) in 2013 or 2014 (n = 97,760) with and without preexisting SMI.

Characteristic	Women with SMI (n = 1581)	Women without SMI (n = 96,179)
	Mean (±SD) or n (%)	
Demographic and socioeconomic characteristics		
Age	60.84 (±11.92)	60.68 (±13.74)
Inclusion in the scheme covering healthcare costs for low-income groups (CMU-C) *	70 (4.43)	3256 (3.39)
Missing values	1 (0.06)	16 (0.02)
Inclusion in the scheme providing allowances for purchasing complementary health insurance (ACS) *	176 (11.13)	2124 (2.21)
Missing values	1 (0.06)	16 (0.02)
Quintile of community-level deprivation index (FDep) *		
1st quintile (least deprived)	277 (17.52)	19,159 (19.92)
2nd quintile	267 (16.89)	18,657 (19.40)
3rd quintile	403 (25.49)	19,655 (20.44)
4th quintile	314 (19.86)	18,396 (19.13)
5th quintile (most deprived)	267 (16.89)	17,170 (17.85)
Missing values	53 (3.35)	3142 (3.27)
Residency in an overseas territory	29 (1.83)	1664 (1.73)
Missing values	0 (0.00)	68 (0.07)
Clinical characteristics		
*Non-metastatic invasive breast cancer*		
Invasive carcinoma with lymph node involvement	293 (18.53)	16,306 (16.95)
Invasive carcinoma without lymph node involvement	1092 (69.07)	67,658 (70.35)
*Others*		
Ductal carcinoma in situ *	37 (2.34)	3109 (3.23)
Lobular carcinoma in situ	4 (0.25)	345 (0.36)
Carcinoma in situ of uncertain type	18 (1.14)	1296 (1.35)
Metastatic breast cancer *	126 (7.97)	6403 (6.66)
Tumor of uncertain malignant potential	11 (0.70)	1062 (1.10)
Comorbidity index *	0.58 (±0.97)	0.31 (±0.70)

Note: Demographic and socioeconomic characteristics reported for the year prior to breast cancer incidence consideration; * *p* < 0.05 for the difference between the SMI and no SMI groups (but the sample is exhaustive).

**Table 2 jcm-12-00412-t002:** Breast cancer care received by women with and without SMI.

	Women with NMI Breast Cancer with Lymph Node Involvement		Women with NMI Breast Cancer without Lymph Node Involvement		Total of Women with NMI Breast Cancer			
	SMI (N = 263)	No SMI (N = 789)	SMI (N = 1083)	No SMI (N = 3249)	Univariate Analysis		Multivariable Analysis ^+^	
	% ^				OR (SMI vs. No SMI)	95% CI	AOR (SMI vs. No SMI)	95% CI
**Main diagnostic tests** ^X^								
Mammogram	91.25	97.72	92.43	97.26	0.32 *	0.24–0.42	0.38 *	0.28–0.51
Breast biopsy	81.37	90.49	83.84	90.67	0.52 *	0.44–0.62	0.56 *	0.47–0.67
Echography	89.73	96.32	87.17	92.67	0.50 *	0.41–0.62	0.58 *	0.47–0.72
MRI	27.76	36.25	23.27	29.70	0.71 *	0.62–0.81	0.76 *	0.67–0.87
Number of main diagnostic tests ^	2.90 (0.96)	3.21 (0.70)	2.87 (0.92)	3.10 (0.72)	0.92 *	0.90–0.94	0.93 *	0.92–0.95
Mammogram + breast biopsy (recommended combination)	79.09	89.48	81.26	89.54	0.49 *	0.42–0.58	0.54 *	0.46–0.64
**Treatment**								
Neoadjuvant chemotherapy	16.73	12.29	6.09	6.09	1.13	0.91–1.40	1.22	0.97–1.54
Lumpectomy	53.99	68.57	72.95	76.58	0.75 *	0.66–0.85	0.82 *	0.70–0.95
Mastectomy	54.75	37.26	26.04	22.19	1.38 *	1.21–1.57	1.38 *	1.20–1.58
Adjuvant chemotherapy	58.94	66.03	21.79	26.75	0.78 *	0.69–0.88	0.80 *	0.70–0.91
Adjuvant radiotherapy	88.59	93.79	74.33	77.75	0.80 *	0.69–0.92	0.87 *	0.75–0.98
Hormone therapy	74.52	82.51	69.99	71.13	0.88	0.77–1.01	0.86 *	0.75–0.99
**Type of treatment combination**								
Operative only	4.18	0.25	7.39	5.48	1.55 *	1.19–2.02	1.47 *	1.13–1.91
Operative + radiotherapy	3.42	1.14	12.28	11.97	1.08	0.88–1.32	1.14	0.93–1.41
Of which lumpectomy only + radiotherapy	0.76	0.89	11.54	11.45	1.01	0.81–1.24	1.08	0.87–1.34
Operative + radiotherapy + chemo/hormone therapy	84.41	91.38	61.03	65.28	0.80 *	0.71–0.91	0.84 *	0.74–0.96
Operative + chemo/hormone therapy	5.70	4.94	11.08	11.30	1.00	0.81–1.22	0.94	0.77–1.16
No operation + any other form of treatment	2.28	2.28	8.22	5.97	1.37 *	1.09–1.72	1.06	0.61–1.81

^x^ Including both hospital and community-based diagnostic tests. ^ Mean (±SD) for number of main diagnostic tests. * *p* < 0.05. ^+^ Adjusted for CMU-C/ACS status, FDep quintile at the place of residence, MRMI synthetic comorbidity index, and type of hospital where first breast cancer treatment was received (this latest explanatory variable was considered for treatment indicators only)—whenever significant at the 0.10 significance level in univariate analyses.

**Table 3 jcm-12-00412-t003:** Indicators of the quality of breast cancer care pathways for women with and without SMI.

Type of Breast Cancer	Indicator of Cancer Care Quality	SMI% (n/d)	No SMI% (n/d)	Target Threshold	Alert Threshold	Univariate Analysis		Multivariable Analysis ^+^	
						OR (SMI vs. No SMI)	95% CI	AOR (SMI vs. No SMI)	95% CI
*Indicators of diagnosis (process)*									
Proportion of women undergoing their first treatment within 6 weeks post-mammogram (Indicator 1) ^#^		39.1 (422/1079)	46.5 (1659/3565)	≥90%	<80%	0.74 *	0.64–0.84	0.75 *	0.65–0.87
Proportion of women undergoing biopsy within 2 weeks post-mammogram (Indicator 2) ^#^		62.7 (677/1079)	66.1 (2355/3565)	≥90%	<80%	0.87 *	0.75–0.99	0.91	0.79–1.06
Proportion of women undergoing their first treatment within 4 weeks post-biopsy (Indicator 3) ^#^		34.8 (376/1079)	41.5 (1479/3565)	≥90%	<80%	0.75 *	0.65–0.86	0.78 *	0.67–0.91
Proportion of women undergoing biopsy prior to first treatment (Indicator 4) ^##^		87.8 (1080/1230)	92.1 (3580/3886)	>98%	≤95%	0.61 *	0.50–0.75	0.65 *	0.52–0.80
*Indicators of diagnosis and treatment (process)*									
Proportion of women without lymph node involvement undergoing sentinel lymph node excision without axillary dissection (Indicator 5)		24.3 (263/1083)	26.2 (849/3245)	>95%	<90%	0.91	0.77–1.06	0.82 *	0.67–0.99
*Indicators of treatment (process)*									
Proportion of women undergoing radiotherapy after breast-conserving surgery (Indicator 6)		91.8 (780/850)	94.2 (2676/2840)	>95%	<90%	0.68 *	0.51–0.91	0.79	0.58–1.09
Proportion of women undergoing adjuvant radiotherapy within 12 weeks post-surgery (Indicator 7A)		82.1 (536/653)	85.0 (1659/1952)	>95%	<90%	0.82	0.65–1.03	0.88	0.69–1.11
Proportion of women undergoing adjuvant chemotherapy within 6 weeks post-surgery (Indicator 7B)		47.3 (185/391)	50.6 (703/1390)	>90%	<85%	0.87	0.70–1.09	0.86	0.69–1.08
Proportion of women undergoing radiotherapy within 6 weeks post-adjuvant chemotherapy (Indicator 8)		74.9 (250/334)	77.5 (907/1171)	>95%	<90%	0.86	0.65–1.13	0.90	0.68–1.20
*Indicators of treatment (outcomes)*									
Proportion of women not treated with neoadjuvant chemotherapy undergoing breast reintervention (Indicator 9)		17.0 (149/875)	14.2 (405/2848)	<10%	>20%	1.24 *	1.01–1.52	1.22	0.99–1.49
*Indicators of follow-up (process)*									
Proportion of women who have had their first follow-up mammogram (Indicator 10)		52.5 (604/1151)	58.5 (2155/3682)	>98%	<95%	0.78 *	0.69–0.89	0.81 *	0.71–0.93

^#^ Calculated among women who received a mammogram and a breast biopsy. ^##^ Calculated among women who received a mammogram. * *p* < 0.05. ^+^ Adjusted for CMU-C/ACS status, FDep quintile at the place of residence, MRMI synthetic comorbidity index, and type of hospital where first breast cancer treatment was received—whenever significant at the 0.10 significance level in univariate analyses. n: numerator, d: denominator.

**Table 4 jcm-12-00412-t004:** Risk of breast cancer specific mortality among women with and without SMI.

Univariate Analysis		Main Multivariable Analysis ^+^		Sensitivity Analyses ^+^			
				(1)		(2)	
HR (SMI vs. No SMI)	95% CI	aHR (SMI vs. No SMI)	95% CI	aHR (SMI vs. No SMI)	95% CI	aHR (SMI vs. No SMI)	95% CI
1.68	1.25–2.24	1.39	1.03–1.87	1.46	1.0–1.94	1.50	1.14–1.99

^+^ Adjusted for MRMI synthetic comorbidity index and type of hospital where the first breast cancer treatment was received; they were significant at the 0.10 significance level in univariate analyses. (1) Considering deaths from any cancer site as the primary event. (2) Considering also deaths with breast cancer recorded as an associated cause of death among primary events.

## Data Availability

The authors are restricted from sharing the data underlying this study because of restrictions that apply to the SNDS database, according to French law and in order to protect health data privacy. Only aggregated data can be made available upon request from the corresponding author.

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
