# Peer review of "Breast Cancer Care Pathways for Women with Preexisting Severe Mental Disorders: Evidence of Disparities in France?"

_jcm, 2023, doi:10.3390/jcm12020412_

Round 1

Reviewer 1 Report

This study evaluated mortality in people with breast cancer with and without severe mental illness, which brings into discussion the disparity in the care pathways received by the group of patients with cancer and severe mental illness. This is a very interesting study, but I consider some issues that could be useful for discussion:

1. The influence of caregivers on the timely care of patients,

2. The impact of family support, 

3. As well as the impact of adherence to pharmacological medication, which is an important issue to consider when dealing with a patient with severe mental illness.

Author Response

First of all, we are very grateful to the reviewers for the time spent reviewing our manuscript and for the constructive criticism, which has helped us improve our manuscript and that we address below point-by-point.

Reviewer 1

This study evaluated mortality in people with breast cancer with and without severe mental illness, which brings into discussion the disparity in the care pathways received by the group of patients with cancer and severe mental illness. This is a very interesting study, but I consider some issues that could be useful for discussion:

  1. The influence of caregivers on the timely care of patients,
  2. The impact of family support, 
  3. As well as the impact of adherence to pharmacological medication, which is an important issue to consider when dealing with a patient with severe mental illness.

Answer: We fully agree with the reviewer that these three points are likely to play a strong role in cancer care pathways, and we thank the reviewer for underscoring it. Unfortunately, these points can not be studied in the SNDS data used in our research which do not include information on caregivers, family support nor adherence to pharmacological medication (just delivery of pharmaceuticals). However, we agree that this should have been discussed in more depth and a paragraph was added in this regard (see page 15, lines 513-523, in track changes mode): “For instance, the data do not include any indicator of social support or marital status, which are likely to play a role in cancer care pathways. The presence of caregivers or strong family support can indeed be expected to facilitate the timely care of patients in comparison to those who are secluded. Their role has been little addressed in the literature on SMI and cancer so far [71], and the qualitative component of our research project, which includes interviews with relatives and caregivers of patients with SMI, may shed additional light on this issue. Similarly, the data do not provide any information on SMI symptom levels and functioning, nor on adherence to pharmacological medications – that can be particularly complicated for psychotropic drugs [72], while there may be specific challenges associated with the care of patients with unstable or untreated psychiatric disorders compared to patients with stabilized SMI [33].”.

Reviewer 2 Report

This study investigated the relationship between women with preexisting SMI and breath cancer care pathway. This is a well-designed and executed experiment. The control group is properly chosen to match the experimental group. Statistical methods are used properly and results are clear and straightforward. 
I just have two minor suggestions:

1) Please remove the "aHR=1.39; 95%CI: 24 1.03-1.87"  in the abstract, as this is justified in the result.

2) Some grammatical errors appeared throughout the manuscript. Please check and correct them.

Author Response

First of all, we are very grateful to the reviewers for the time spent reviewing our manuscript and for the constructive criticism, which has helped us improve our manuscript and that we address below point-by-point.

This study investigated the relationship between women with preexisting SMI and breath cancer care pathway. This is a well-designed and executed experiment. The control group is properly chosen to match the experimental group. Statistical methods are used properly, and results are clear and straightforward. 

I just have two minor suggestions:

1) Please remove the "aHR=1.39; 95%CI: 24 1.03-1.87" in the abstract, as this is justified in the result.

Answer: Thank you for pointing this out, we agree that these figures should be interpreted considering all associated methodological details and sensitivity analyses, and so they were removed from the abstract as suggested (see abstract, in track changes mode).

2) Some grammatical errors appeared throughout the manuscript. Please check and correct them.

Answer: Thank you for pointing this out as well. We thoroughly proof-read our manuscript to try and identify all remaining grammatical errors (in track changes modes throughout the manuscript). However, if you feel that we failed to address some of those you noticed, please do not hesitate to let us know, as we are keen to make our manuscript as readable as possible.
